# Excretory/Secretory Products from *Schistosoma japonicum* Eggs Alleviate Ovalbumin-Induced Allergic Airway Inflammation

**Zhidan Li**[1,2], **Xiaoling Wang**[3], **Wei Zhang**[3], **Wenbin Yang**[3], **Bin Xu**[2], **Wei Hu**[2,3,4,5] *

**1** Department of Immunology, Binzhou Medical University, Yantai, Shandong, P. R. China, **2** National Institute of Parasitic Diseases, Chinese Centre for Disease Control and Prevention, WHO Collaborating Centre for Tropical Diseases, National Centre for International Research on Tropical Diseases, Key Laboratory of Parasite and Vector Biology of the Chinese Ministry of Health, Shanghai, China, **3** State Key Laboratory of Genetic Engineering, Ministry of Education Key Laboratory of Contemporary Anthropology, Human Phenome Institute, Ministry of Education Key Laboratory for Biodiversity Science and Ecological Engineering, Department of Microbiology and Microbial Engineering, School of Life Sciences, Fudan University, Shanghai, China, **4** Department of Infectious Diseases, Huashan Hospital, Fudan University, Shanghai, China, **5** College of Life Sciences, Inner Mongolia University, Hohhot, China

* huw@fudan.edu.cn

**Data Availability Statement:** The raw data supporting the conclusions of this article will be made available by the authors, without undue reservation.

## Abstract

### Introduction

Excretory/secretory products (ESPs) derived from helminths have been reported to effectively control allergic inflammation, which have better therapeutic prospects than live parasite infections. However, it remains unknown whether ESPs from schistosome eggs can protect against allergies, despite reports alleging that schistosome infection could alleviate disordered allergic inflammation.

### Method

In the present study, we investigated the protective effects of ESPs from *Schistosoma japonicum* eggs (ESP-SJE) on asthmatic inflammation. Firstly, we successfully established an allergic airway inflammation model in mice by alum-adjuvanted ovalbumin (OVA) sensitization and challenge. ESP-SJE were administered intraperitoneally on days -1 and 13 (before sensitization), on day 20 (before challenge), and on days 21–24 (challenge phase).

### Results

The results showed that ESP-SJE treatment significantly reduced the infiltration of inflammatory cells, especially eosinophils into the lung tissue, inhibited the production of the total and OVA-specific IgE during OVA-sensitized and -challenged phases, respectively, and suppressed the secretion of Th2-type inflammatory cytokines (IL-4). Additionally, ESP-SJE treatment significantly upregulated the regulatory T cells (Tregs) in the lung tissue during OVA challenge. Furthermore, using liquid chromatography-mass spectrometry analysis and Treg induction experiments *in vitro*, we might identify nine potential therapeutic proteins

**Funding:** This work has been supported by funding from the National Natural Science Foundation of China (31725025 to WH), Shanghai Municipal Science and Technology Committee of Shanghai outstanding academic leaders plan (18XD1400400 to WH), Science and Technology Leading Talent Team in Inner Mongolia Autonomous Region (2022LJRC0009 to WH) of Wei Hu, who was this MS's corresponding author. The funder WH participates in study design, data collection and analysis, decision to publish, or preparation of the manuscript.

**Competing interests:** The authors have declared that no competing interests exist.

against allergic inflammation in ESP-SJE. The targets of these candidate proteins included glutathione S-transferase, egg protein CP422 precursor, tubulin alpha-2/alpha-4 chain, actin-2, T-complex protein 1 subunit beta, histone $H_4$, whey acidic protein core region, and molecular chaperone HtpG.

## Conclusion

Taken together, the results discussed herein demonstrated that ESP-SJE could significantly alleviate OVA-induced asthmatic inflammation in a murine model, which might be mediated by the upregulation of Treg in lung tissues that may be induced by the potential modulatory proteins. Therefore, potential proteins in ESP-SJE might be the best candidates to be tested for therapeutic application of asthma, thus pointing out to a possible new therapy for allergic airway inflammation.

## Author summary

Asthma, especially atopic asthma, one of the most common chronic, non-communicable diseases in children and adults, which is characterized by variable respiratory symptoms and variable airflow limitation. As the rapid rising of allergy and asthma rates, it seriously affects the quality of people's life. Asthma is a consequence of complex gene-environment interactions with heterogeneity in clinical presentation. Recent years, A leading theory is the "hygiene hypothesis", which suggests that the decreasing incidence of infections in western countries is the origin of the increasing incidence of allergic diseases. Following this lead, abundant experimental studies have proved that helminth infection can down-regulate host's immunopathology in allergies. Further, helminth-derived excretory/secretory products (ESPs) have been reported to effectively control allergic inflammation, which have better therapeutic prospects than live parasite infections. Schistosome act as a wide-spread and one of the best illustrated helminths. Present study investigated that ESPs from *Schistosoma japonicum* eggs (ESP-SJE) could effectively alleviate OVA-induced allergic airway inflammation in a murine model, and nine potential modulatory proteins for Treg induction that might to be vital to inhibit asthmatic inflammation from ESP-SJE were predicated by mass spectrum analysis. The significance of our research is in founding a new way to explore schistosome-derived molecules alleviating asthma, which may allow the development of a model that will greatly enhance our understanding of the interaction between atopic asthma and helminth parasites, further facilitate the discovery of a better candidates to be tested for therapeutic application of asthma, thus pointing out to a possible new therapy for allergic disease or other autoimmune disease.

## Introduction

The hygiene hypothesis proposes that the increased prevalence of asthma is linked to reduced exposure to infectious agents, such as parasites, in early childhood [1–3]. In the absence of parasitic infections, the immune system becomes more active, which increases the occurrence of immune-mediated diseases such as allergic disorders and autoimmune diseases [4, 5]. Since there is no effective and safe treatment for these allergic conditions, studies on parasite-derived

regulatory molecules have been gaining increasing attention, whose promising results could pave the way for controlling allergic diseases.

It is known that allergic diseases affect approximately 30% of the global population, and asthma is a major public health issue, causing pain and suffering to an estimated 300 million individuals worldwide [6, 7], a large proportion of which is constituted by children [8, 9]. Allergic asthma is a chronic inflammatory airway disease characterized by the infiltration of T-helper cell (Th) type 2 and eosinophils into the airway wall. Th2 cytokines (namely, interleukin (IL)-4, IL-5 and IL-13) are important molecules involved in the emergence of asthma, contributing to features of the disease, including immunoglobulin (Ig) E production, eosinophil accumulation in the lungs, and mucus hypersecretion and airway hyper reactivity (AHR) [10, 11]. Furthermore, it has been demonstrated that regulatory T cell (Treg) defective function and disordered suppressive activity, as well as the overactivation of Th2 effector cells, are the major inducers of allergic disorders [12, 13]. Corticosteroids can effectively control allergy inflammation, but their use is associated with severe long-term side effects, and certain patients do not respond to treatment with corticosteroids [14–17].

Helminth therapy was firstly proposed in the 1990s using ova from *Trichuris suis*, and subsequent studies have demonstrated that various helminthic species, including *T. suis*, *Nippostrongylus brasiliensis*, *Schistosoma* spp., and others, can effectively alleviate allergic airway inflammation (AAI) in experimental allergic asthma models and early clinical trials [18,19]. Schistosomes are among the helminths which confer significant protective effect against allergic sensitization and challenge [20–23]. It has been reported that AAI was effectively inhibited 3–5 weeks after infection with *Schistosoma japonicum* in a murine model [21]. Moreover, it has been shown that the protective effect of *Schistosoma mansoni* infection on AAI mainly depends on the chronicity of infection [23]. Thus, it can be suggested that the immunoregulatory roles of schistosome infection on AAI mainly occur in the chronic phase of infection, during which egg and egg-derived products closely interact with the host immune response [24]. Despite the significant benefits of probiotic worm therapy in the treatment of chronic inflammatory conditions, concerns persist around the implications of experimental human infection with a live pathogen and its potential scalability, such as the pathological damage induced by infection [25, 26]. Thus, identifying worm fractions or molecules with protective effects and promoting the development of helminth-derived molecules for treating disorders resulting from a dysregulated function of the immune system can be considered a promising therapeutic approach.

Excretory/secretory products (ESPs) are cell-released substances (mostly proteins) involved in various biological processes, such as cell-cell communication, cell migration, signal transduction, adhesion, invasion, and potential infective strategies in disease mechanisms [27, 28]. ESPs from various helminth parasites have been reported to suppress lung inflammation in mouse models [29–31]. Verissimo et al. found that ESPs derived from *S. japonicum* eggs (ESP-SJEs) can modulate the immune system in hosts by eliciting the secretion of interleukins related to the Th2 response [32]. More recently, Gong et al. demonstrated that ESP-SJE antigens could drive macrophage polarization into an immunoregulatory M2b phenotype [33]. Furthermore, other studies showed that antigens and other products from schistosome eggs had a well regulatory role against the host, as demonstrated by a reduced but still predominant Th2 response due to a persistent Treg environment [34–37]. Collectively, the above studies suggested that ESPs derived from *S. japonicum* eggs play important immunomodulatory roles in host-pathogen interactions. However, it remains unclear whether ESP-SJE can suppress hyperactivity during AAI.

Thus, the aim of this study was to investigate the immuno-protective effect of ESP-SJEs against AAI in a murine model. Moreover, the identification of specific fractions of

ESP-SJEs that may be useful in relieving alum-adjuvanted ovalbumin (OVA)-induced AAI was conducted. Furthermore, Treg induction experiments *in vitro* and liquid chromatography-mass spectrometry (LC-MS) analysis were performed to further identify potential therapeutic proteins isolated from ESP-SJE. Taken together, the results discussed herein may be useful for the development of novel therapeutic agents for important human diseases.

## Materials and methods

### Ethics statement

This study was performed in strict accordance with the recommendations in the Guide for the Care and Use of Laboratory Animals at National Institute of Parasitic Disease, Chinese Center for Disease Control and Prevention (NIPD/China CDC). All procedures on animal experiment complied with the guideline of the Laboratory Animal Welfare & Ethic Committee (LAWEC) of National Institute of Parasitic Diseases (Permit Number: IPD-2016-7). All efforts were made to minimize suffering of the animals.

### Animals

Six- to eight-week-old female BALB/C mice were purchased from the Shanghai SLAC Laboratory Animal Center (Shanghai, China) (permit number: SCXKZ Shanghai 2018–0004). Animals were reared in pathogen-free animal houses of the National Institute of Parasitic Disease (NIPD), Chinese Center for Disease Control and Prevention, located in Shanghai, China. Experimental rabbits were obtained from Shanghai JieSiJie Laboratory Animal Co., and reared in clean animal houses of NIPD. All animals were provided with access to abundant sterilized water and food at appropriate environmental conditions of temperature (22–26˚C) and humidity (50%–60%).

### Isolation of ESPs from S. japonicum eggs

Female rabbits were infected with 800–1000 cercariae via abdominal skin penetration, after 6 weeks, the rabbits were euthanized and liver tissues were excised and *S. japonicum* eggs were isolated from liver tissues as described previously [38]. Liver tissue fragments were minced and then stirred at low temperatures to obtain a homogenate. The homogenate supernatant was passed through sterile 80-, 120-, and 150-micron mesh filters, washed several times with cold phosphate buffer solution (PBS), and then centrifuged at $450 \times g$ for 10 min. Subsequently, the suspension of eggs and cells was submitted to digestion with 1 mg/mL collagenase IV (Cat# C8160, Solarbio, China) and 0.2 mg/mL DNase (Cat# AMDP1; Sigma-Aldrich, St. Louis, MI, USA) in a $CO_2$ incubator at 37˚C for 45 min, and then centrifuged at $300 \times g$ for 10 min. Purified schistosome eggs were collected and cultured in free Roswell Park Memorial Institute (RPMI) medium (Cat# PM150110, Procell, China) supplemented with 100 U/mL penicillin and 100 μg/mL streptomycin (Cat# PB80120, Procell, China). After 48 h, culture supernatants were harvested and centrifuged at $200 \times g$ for 10 min to remove eggs, then at $10,000 \times g$ for 10 min to remove debris. Purified ESPs products were obtained, and their concentration was determined using a Bradford protein assay. Endotoxin level of ESPs antigens was <0.03 EU/mL, as determined by a Limulus amoebocyte lysate assay (Cat#: L00350C; Genscript, China) according to the manufacturer's instructions. Finally, ESPs products were stored at -80˚C until further use.

## Establishment of an OVA-induced AAI model

BALB/C mice were randomly divided into five experimental groups (containing at least six mice per group): i) OVA-induced AAI (OVA group); ii) OVA-induced AAI treated with ESP-SJE (OVA+E/S group); iii) OVA-induced AAI treated with dexamethasone (DXM) (OVA+DXM group), iv) treatment with ESP-SJEs only (E/S group); and v) normal control without any treatment (NOR group). The first day of sensitization was considered day 0. On day 0 and day 14, 10 μg of OVA (Cat# 77120; Sigma-Aldrich, USA) and 2 mg of aluminum hydroxide gel were dissolved into 0.1 mL PBS and intraperitoneally injected into mice to induce sensitization. Then, on days 21–24, mice in the OVA-induced AAI model were challenged with exposure to 1% OVA aerosols (prepared in PBS) for 30 min every day in an airtight chamber using a Medical Compressor Nebulizer (DEDAKJ, Germany) (Fig 1A and 1B). Mice in the NOR group and the E/S group were submitted to exposure to aerosols with the same volume of PBS alone. Treatment with ESP-SJEs (1 mg per mouse per time) was administered on days -1, 13, and 20–24, for a total of seven times. All animals were euthanized 48 h after the last challenge with OVA aerosol (day 26).

## Culture of bronchoalveolar lavage fluids (BALFs) and cell count determination

BALF was collected from mice of all five experimental groups 48 h after the last challenge with OVA aerosol. Firstly, a tracheotomy was performed, and an arteriovenous indwelling needle (Cat# 4254112B; 20G; BRAUN, Germany) was inserted into the trachea. Pulmonary alveoli were carefully washed twice with 0.3 mL of cold PBS, and 0.5–0.6 mL of BALFs were collected by centrifugation at 3,000 rpm for 10 min at 4°C. The supernatants were stored at -80°C for further determination of cytokines. Cell pellets were fixed in 1 mL of 4% paraformaldehyde and submitted to hematoxylin-eosin (H&E) staining. The number of total cells, eosinophils, macrophages, neutrophils, and lymphocytes was determined in a high-resolution inverted microscope (NIKON, Japan). One thousand cells from multiple fields of view were examined for each slide preparation to determine the counts of eosinophils, macrophage, neutrophils, and lymphocytes on blinded samples [39].

## Lung histopathology

Lung tissues from mice of all five groups were fixed overnight in 4% phosphate buffered formaldehyde, then embedded in paraffin, and cut into 5-mm sections for H&E staining and periodic acid-Schiff (PAS) staining according to standard protocols. The images of stained lung tissue sections were visualized in a NIKON DS-U3 microscope (NIKON, Japan). Goblet cell metaplasia intensity and bronchial and peribronchovascular inflammation were assessed by H&E and PAS staining, respectively, and 0–4 scores were assigned to samples by two blinded, independent investigators, as described previously [40].

## Detection of OVA-specific IgE, IgG1 and IgG2a antibodies in serum

Total IgE levels and OVA-specific IgE, IgG1 and IgG2a levels were measured using an enzyme linked immunosorbent assay (ELISA) according to the manufacturer's instructions. Maxisorp 96-well microliter plates (Thermo Fisher Scientific, USA) were coated with 1 μg/mL of rat monoclonal anti-mouse IgE antibody for measuring total IgE levels (Cat# ab99571; 1: 1000; Abcam, UK) or with 10 μg/mL of ovalbumin for measuring OVA specific IgE, IgG1 and IgG2a levels (Cat# A5503; Sigma, USA) in 100 μL of carbonate-bicarbonate buffer (pH 9.6) at 4°C for 12–16 h. Then, plates were blocked with 100 μL/well of PBS with 0.05% Tween 20 (PBST)-

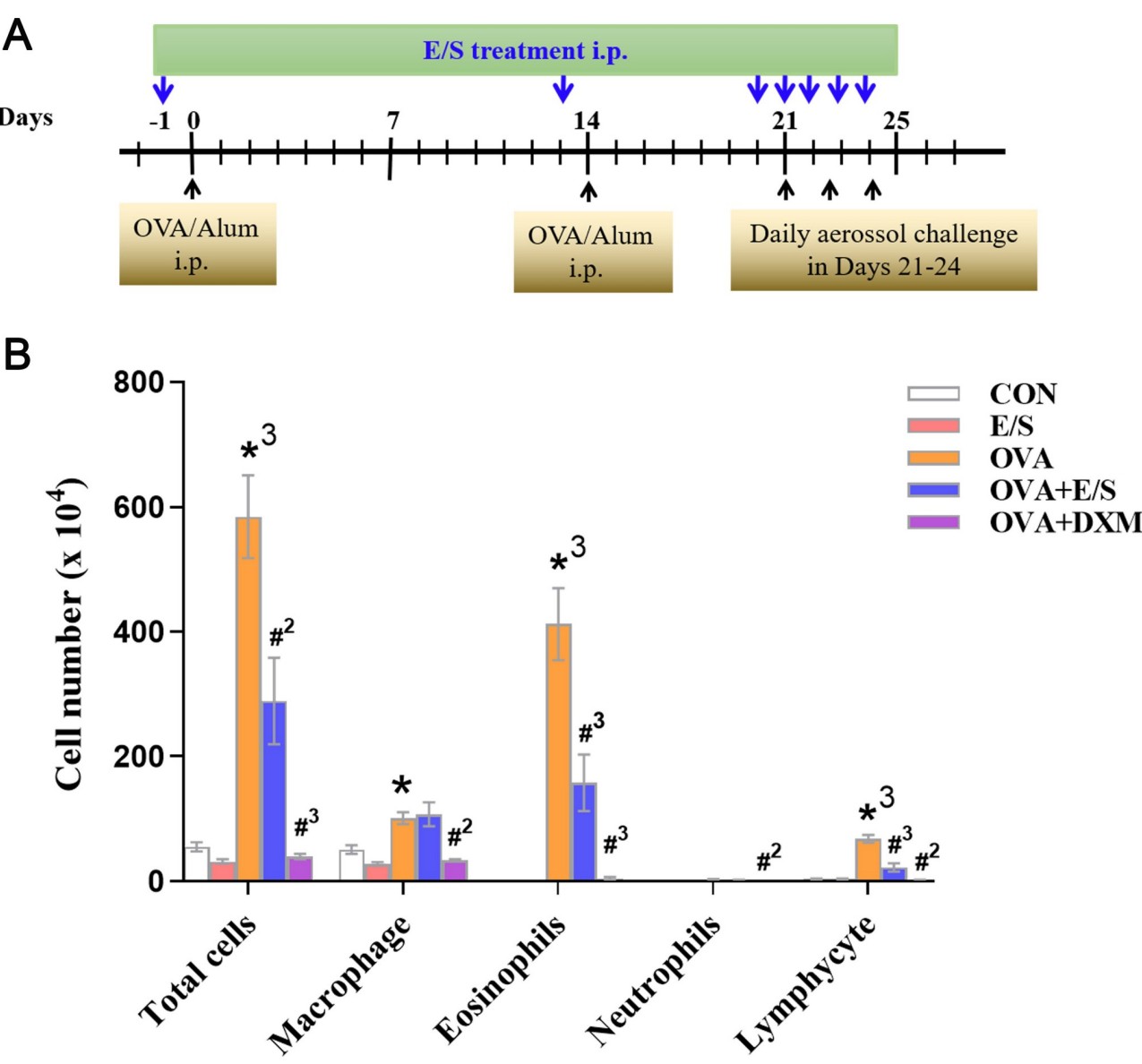

**Fig 1. ESP-SJE reduced OVA induced inflammatory cell counts in BALFs.** (A) Experimental design of OVA induced AAI treated with ESP-SJE in murine model. (B) Inflammatory cell (Total cell, Macrophage, Eosinophils, Lymphocytes, Neutrophils) infiltration in BALF of mice after OVA challenge were compared in CON, E/S, OVA, OVA+E/S and OVA+DXM groups. CON, normal mice (without OVA sensitization and challenge); E/S, mice without OVA sensitization and challenge but treated with ESP-SJE; OVA, mice with OVA sensitization and challenge but without ESP-SJE treatment; OVA + E/S, mice sensitized and challenged with OVA and treated with ESP-SJE; OVA + DXM, mice sensitized and challenged with OVA and treated with dexamethasone. All data were shown as mean ± SEM and analyzed by the one-way analysis of variance (ANOVA) with Tukey test. At least 5 mice per group, experiment performed twice. *, *[2], *[3] indicates $P < 0.05$, $P < 0.01$ and $P < 0.001$, respectively, OVA group versus CON group; #, #[2] and #[3] indicates $P < 0.05$, $P < 0.01$ and $P < 0.001$, respectively, OVA+E/S group or OVA+DXM group versus OVA group.

Bull Serum Albumin (BSA) for 2 h at 37˚C. After 5–6 rounds of washing with PBST, 100 μL of serum dilutions (diluted with PBST at a ratio of 1:40 for total IgE, and at a ratio of 1:5 for OVA-specific IgE, IgG1 and IgG2a) were added to each well, followed by incubation for another 2 h at 37˚C. Subsequently, 100 μL of HRP-labeled goat anti-mouse IgE, IgG1 or IgG2a (Cat# ab99574, ab97240, and ab97245; 1: 2000; Abcam, UK) antibodies previously diluted with

PBST were added to each well. After incubation for 2 h at 37˚C, plates were washed with PBST for 5–6 times. The color reaction was evaluated by adding 100 μL of 3,3',5,5'-tetramethylbenzidine (TMB) (Cat# PA107; TIANGEN, China) per well, followed by incubation for 30 min at 37˚C. The reaction was then stopped with 5% sulfuric acid (50 μL/well), and $OD_{450}$ values of OVA-specific IgE, IgG1 and IgG2a samples were determined using a multimode microplate reader (BioTek, USA). The determination of total IgE levels was performed using a standard curve.

### Determination of cytokine levels in BALF

The levels of IL-4, IL-5, IL-13, IL-10, eotaxin, IFN-γ, IL-17A, and TNF-α in BALF samples were determined using an 8-plex custom-made Bio-Plex Pro Reagent Kit (Wayen Biotechnologies, China) according to the manufacturer's instructions. The absorbance of samples was determined using a corrected Bio-Plex MAGPIX system (Bio-Rad, Luminex Corporation, Austin, TX, USA), and cytokine levels were calculated using Bio-Plex manager 6.1 (Bio-Rad).

### Preparation of lung tissue suspensions

Lung and spleen tissues were collected from mice of all five groups. After washing with RPMI medium was repeated for 3–4 times, lung tissue was fragmented and digested with 1 mg/mL collagenase type IV (Cat# 8160; Sigma-Aldrich, USA) and 0.2 mg/mL DNase (Cat# AMDP1; Sigma-Aldrich, St. Louis, USA) at 37˚C for 30 min. Digested lung tissue was then filtered through a sterile 48-micron mesh filter and washed twice with cold PBS. Subsequently, mononuclear cells were obtained by Ficoll density gradient centrifugation (Cat# LTS1092; TBD Science, China) and further used in flow cytometry analysis.

Spleen tissues were immediately filtered through a 70-μm cell strainer, washed twice with PBS, and then lysed with a Red Blood Cell Lysis Buffer (Cat# R1010; Solarbio, China). Single cells from spleen tissue were collected and further used in flow cytometry analysis.

### Detection of Treg in lung and spleen tissues

Single cells were incubated with corresponding membrane monoclonal antibodies (mAbs) against FITC-conjugated anti-CD4 (Cat# 88-8111-40; eBioscience, USA) and APC-conjugated anti-CD25 (Cat# 88-8111-40; eBioscience, USA) at 4˚C in the dark for 30 min, and then washed once with cold PBS. Subsequently, cells were fixed and permeabilized according to the manufacturer's instructions using a fixation and permeabilization kit (Cat# 88-8111-40; eBioscience, USA) and then washed with PBS. Then, cells were then stained with a PE-conjugated anti-Foxp3 mAb (Cat# 88-8111-40; eBioscience, USA) at 4˚C for 45 min in the dark. Finally, after being washed twice with PBS, cells were resuspended in 500 μL of PBS and further used for flow cytometry analysis (Cytometer LX, Beckman, USA).

### Preparation of ESP-SJE fractions

ESP-SJEs were collected and separated into different fractions using gel filtration and ion exchange chromatography with an AKATA purifier (GE healthcare, USA). The purifier was prepared in advance, and the liquid flow channel was washed thrice with PBS. The gel column SephadexG100 and the ion Q column were placed in the purifier, and the channel was washed thrice. Collected ESP-SJEs were separated into two groups, F1 and F2, using gel separation, and into four contents, namely, Q1, Q2, Q3, and Q4, by ion exchange. The ESP-SJEs fractions were concentrated to approximately 0.5 mL using a protein ultrafiltration tube (3K; Cat# 88512; Thermo Fisher, USA). The protein concentration of ESP-SJEs fractions was measured

using a Bradford assay. The endotoxin content of ESP-SJEs was found to be below 0.03 EU/mL using a Limulus amoebocyte lysate assay (Cat# L00350C; Genscript, China) according to the manufacturer's instructions. The ESP-SJEs fractions were stored at -80˚C until further use.

### In vitro induction of Treg

*In vitro* induction ability of Treg following treatment with different ESP-SJEs fractions was evaluated. Firstly, single cells collected from spleen tissue of wide-type mice were isolated by grinding and washing the tissue for multiple times. Then, $1*10^6$ cells were placed in each well and treated with α-CD3/CD28 antibodies in RPMI medium containing 10% fetal bovine serum (FBS) and 1% streptomycin and penicillin for 2 h. Subsequently, cells were cultured without or with ESP-SJEs at different work concentrations (namely, 5, 10, and 20 mg/mL) for 3–5 days. Cells were then collected and stained with a commercial kit (Cat# 88-8111-40; eBioscience, USA) for $CD4^+CD25^+FOXP3^+$ Treg. The percentage of Treg in each sample was analyzed by flow cytometry and determined by between-group comparison.

### Tryptic digestion of ESP-SJE for LC-MS/MS analysis

Dried protein pellets of ESP-SJE fractions were suspended in 30 μL of 0.1 M ammonium bicarbonate buffer, and 4 μL of dithiotreitol (45 mM in 0.1 M ammonium bicarbonate buffer) was added to the suspension, followed by incubation at 56˚C for 45 min to reduce alkylation of cysteine residues. After cooling to room temperature, 4 μl of iodoacetamide solution (100 mM iodoacetamide in 0.1 M ammonium bicarbonate buffer) was added to the previous mixture, and incubated in the dark for 30 min. The samples were diluted with 95 μL of 0.1 M ammonium bicarbonate buffer, and 0.1 μg of trypsin was dissolved in 50 mM acetic acid, and maintained at 37˚C for 16–18 h. The reaction was stopped with 2.5 μL of 20% trifluoroacetic acid with water as solvent (*v/v*), and samples were then dried.

### LC-MS/MS analysis and data processing

LC-MS/MS was performed by orbitrap fusion lumos mass spectrometer coupled with an iRT-corrected prediction system. The data were analyzed using a data independent acquisition (DIA) analysis. All samples were analyzed using a single-column system with a manually loaded analysis column (75 μm i.d. ~25 cm; 2.4 μm, ReproSil-Pur120C18-AQ, Dr. Maisch Gmbh, Germany). Library samples were analyzed by mass spectrometry in the data dependent acquisition (DDA) mode. The resolution of primary ion mass spectrometry was 60,000, whereas the resolution of secondary ion mass spectrometry was 15,000, with cycle time set to 3 s in the HCD fragmentation mode. Samples from experimental groups were analyzed using the DIA mode.

DDA data used for library construction of mass spectra were analyzed using Proteome Discoverer (version 2.4; Symefly, Germany) using the Mascot server (version 2.3; Matrix Science, UK). The latest fourth-generation protein library (SjV4) of *S. japonicum* was used to search the mass spectra library. The results were verified based on the Percolator and Mascot values obtained with Proteome Discoverer. Peptides showing false discovery rate (FDR) values and MASCOT *P* values ≤ 0.01 and 0.05 were regarded being confidence.

DIA data analysis was conducted using Skyline software, and the obtained csv file was exported after completing the configuration step by step. Any #N/A entry in the data set was replaced with 0, and data were normalized to determine the expression levels of all identified proteins in each ESP fraction.

## Statistical analysis

All statistical analyses were performed in GraphPad Prism 8.0 (GraphPad Software, Inc., San Diego, CA, USA) using one-way analysis of variance. Data of quantitative variables are shown as mean ± standard error of mean (SEM). Comparative differences were considered statistically significant when *P* values were below 0.05.

# Results

## ESP-SJEs Treatment Suppressed OVA-Induced Inflammatory Eosinophil Recruitment in Lung Tissue

Herein, an OVA-induced AAI murine model was successfully established. On day 0 (first day of OVA sensitization) and day 14, mice were administered OVA intraperitoneally, which was considered the sensitization phase. During days 21–24, mice were exposed daily to 1% OVA aerosols, which was considered the effective phase, as shown in Fig 1A. Intraperitoneal injection of ESP-SJEs was conducted in mice on days -1, 13, and 20–24, mice in the NOR group and ESP-SJEs -treated groups were injected with the same amount of PBS.

Eosinophil infiltration in airway epithelium and lung tissue is an important characteristic of asthmatic inflammation [41]. In the present study, the number of total cells, eosinophils, macrophages, lymphocytes, and neutrophils in BALF of mice was determined by analyzing cell smear between experimental groups. The results showed that OVA sensitization and challenge significantly increased the number of total cells, macrophages, lymphocytes and especially eosinophils ($P < 0.001$) compared to mice in the NOR group. As expected, treatment with ESP-SJEs in the AAI model significantly reduced the number of total cells, lymphocytes, and particularly eosinophils in BALF compared to mice in the OVA group (Fig 1B). However, the reduction in the number of these cell types observed in samples of the DXM group and the OVA group was considerably more significant than that observed in mice in the OVA+E/S group and the OVA group (Fig 1B). Moreover, the number of these cell types in mice treated with ESP-SJE was not significantly different, with only a slight reduction, compared to mice in the NOR group.

## ESP-SJEs Treatment Alleviated OVA-Induced Lung Alterations

To further investigate the protective role of ESP-SJEs on lung inflammation, we performed HE staining for evaluating inflammatory cell infiltration. In addition, PAS staining was conducted for evaluating mucus hypersecretion and goblet hyperplasia in mice lung tissue. The semi-quantitative analysis performed on data retrieved from histological observations indicated that OVA sensitization and challenge induced marked infiltration of inflammatory cells (Fig 2A and 2B) and goblet hyperplasia (Fig 2C and 2D). Furthermore, during AAI, treatment with ESP-SJEs significantly reduced eosinophil-rich leukocyte infiltration and goblet cell hyperplasia compared to OVA-exposed mice. As expected, the anti-inflammatory effect of DXM was considerably higher than that of ESP-SJEs; nonetheless, no significant difference was found between mice treated with ESP-SJEs and those in the NOR group (Fig 2).

## ESP-SJEs Treatment Reduced the Production of OVA-Specific IgE in the Serum

IgE and antibodies are the main factors causing allergic reactions [42]. Therefore, we sought to examine total IgE levels and OVA-specific IgE levels in mice serum on day 20 (sensitization phase) and day 25 (effector phase). Additionally, OVA-specific IgG2a and IgG1 levels were determined on day 25 by ELISA. As shown in Fig 3A and 3B, OVA sensitization significantly

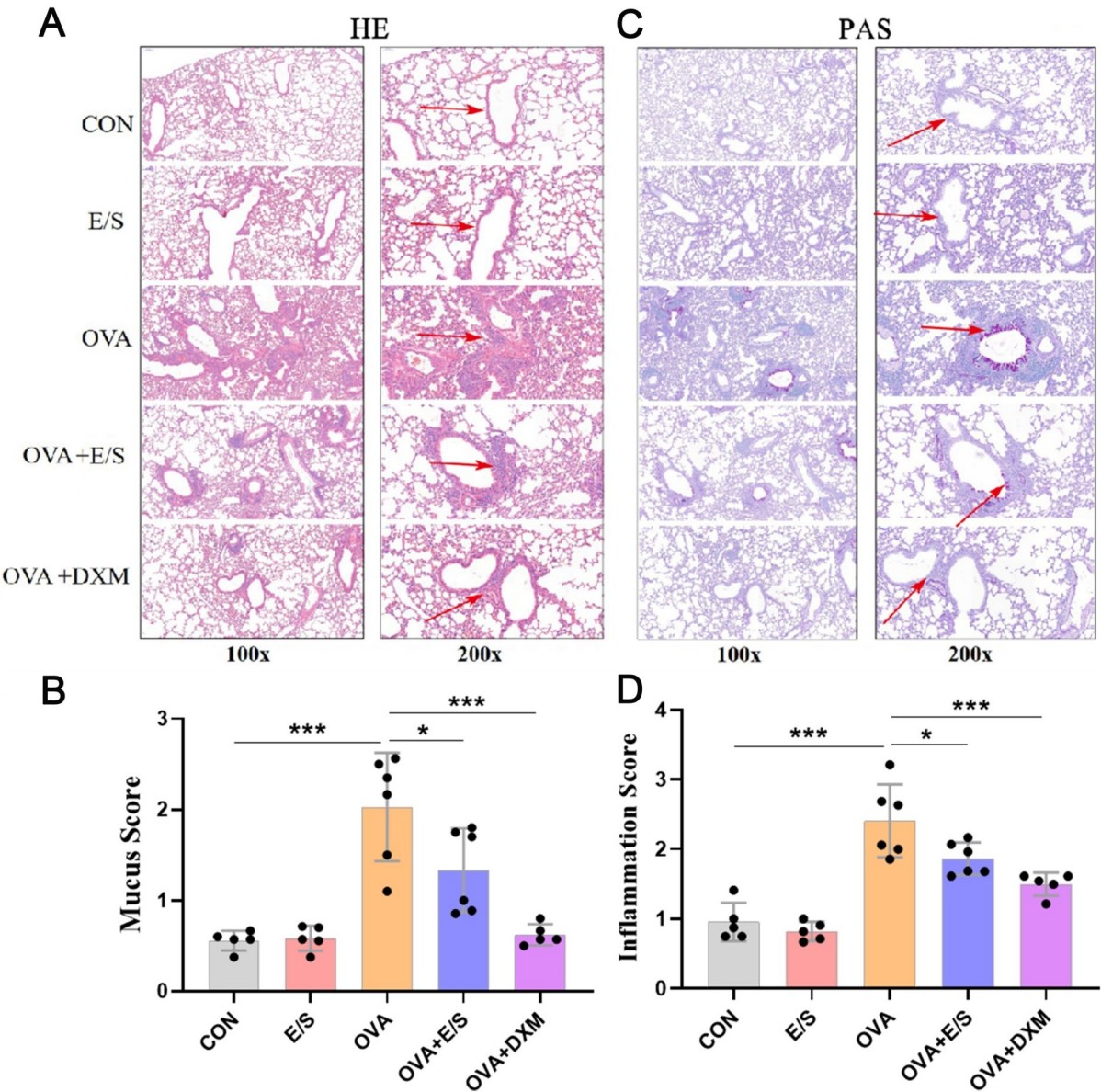

**Fig 2. ESP-SJE alleviates OVA induced lung inflammation and hyperplasia of goblet cell.** (A&B) Representative images of H&E staining of lung tissue and their statistical analysis of inflammation score among CON, E/S, OVA, OVA+E/S and OVA+DXM groups. (C&D) Representative images of PAS staining of lung tissue and their statistical analysis of mucus secretion score among five groups. All data were shown as mean ± SEM. At least 5 mice per group, experiment performed twice. * and *** indicates $P < 0.05$ and $P < 0.001$ by the one-way analysis of variance (ANOVA) with Tukey test.

induced the production of total IgE and OVA-specific IgE on day 20 and day 25. However, after treatment with ESP-SJEs during AAI, the levels of both molecules were reduced to a lower level comparable to that observed in mice of the OVA group, which was also almost comparable to the levels observed in mice of the OVA+DXM group. Furthermore, total IgE levels were significantly higher on day 25 compared to those on day 20 in mice of the OVA, OVA+E/S, and OVA+DXM groups. However, OVA-specific IgE levels were significantly higher on day 20 compared to day 25 in the OVA group (Fig 3A and 3B). Similarly, OVA-

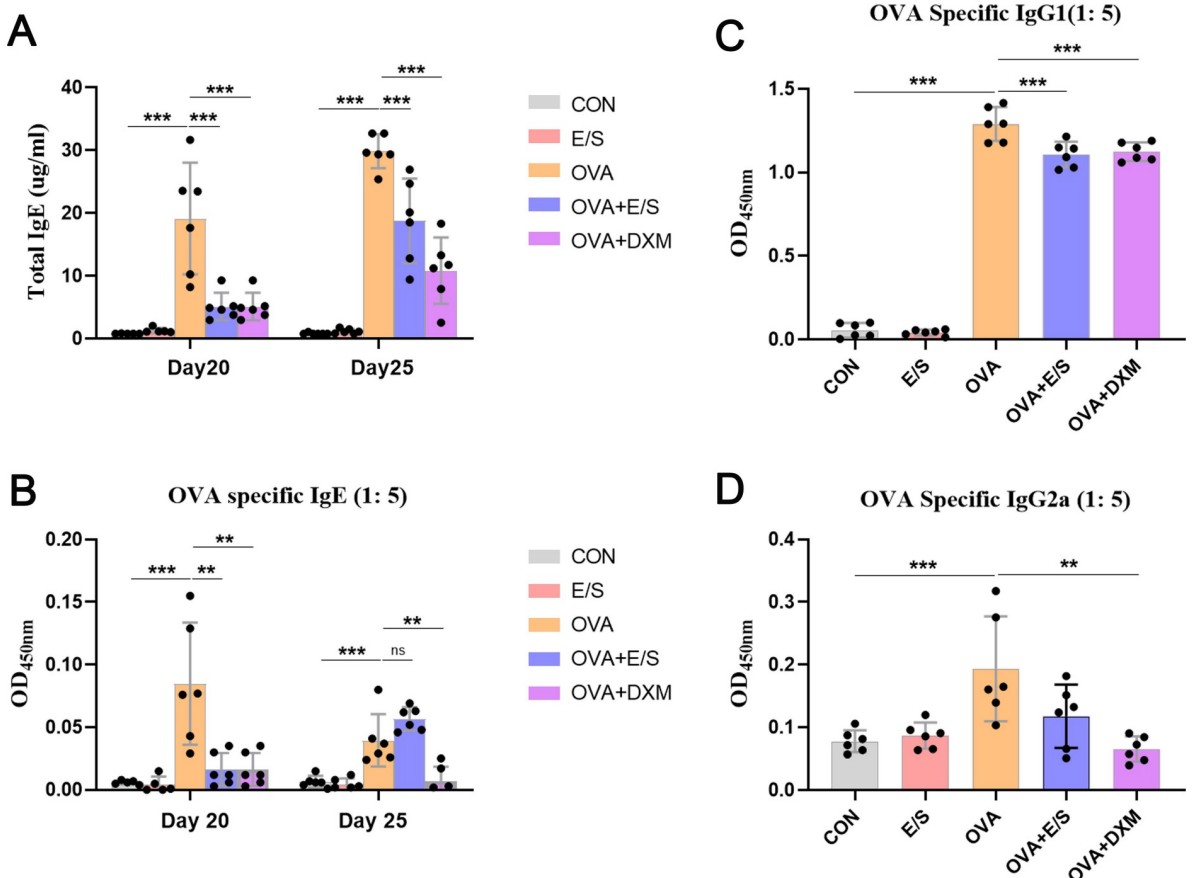

**Fig 3. ESP-SJE treatment inhibit total IgE and OVA specific IgE secretion in murine serum of AAI.** (A) The concentration of total IgE in mouse serum at day 20 (before OVA challenge) and 26 (after OVA challenge) were compared among all groups after OVA challenge in CON, E/S, OVA, OVA+E/S and OVA+DXM groups. (B) OVA specific IgE in sera at day 20 (before OVA challenge) and 26 (after OVA challenge) were measured by ELISA among five groups. OVA specific IgG1 (C) and IgG2a (D) at day 26 were tested by ELISA between groups. All data were shown as mean ± SEM. At least 5 mice per group, experiment performed twice. ** and *** indicates $P < 0.01$ and $P < 0.001$, ns indicates no significant difference, respectively, by the one-way analysis of variance (ANOVA) with Tukey test.

specific IgG1 and IgG2a production was significantly induced on day 25 after OVA sensitization and challenge compared to mice in the NOR group. However, after treatment with ESP-SJEs, OVA-specific IgG1 and IgG2a levels were significantly reduced compared to mice in the OVA group, with only a slight increase compared to mice in the DXM group (Fig 3C and 3D).

## SESP-SJEs Treatment induced treg in lung tissues

Treg cells play a key role in the development of asthma [43]. A previous study showed that *S. mansoni*-mediated suppression of AAI is mediated by infection-induced Treg cells [44]. Herein, the proportion of CD4$^+$CD25$^+$Foxp3$^+$ Treg cells in lung and spleen tissues of mice was determined by flow cytometry. As shown in Fig 4, the proportion of Treg in lung and spleen tissues was slightly reduced only in mice in the OVA group or E/S group, although the observed difference was not statistically significant compared to the normal control. However, compared to the OVA group, the proportion of Treg in lung tissue of mice in the OVA+E/S group was significantly increased, at levels comparable to those found in mice of the DXM

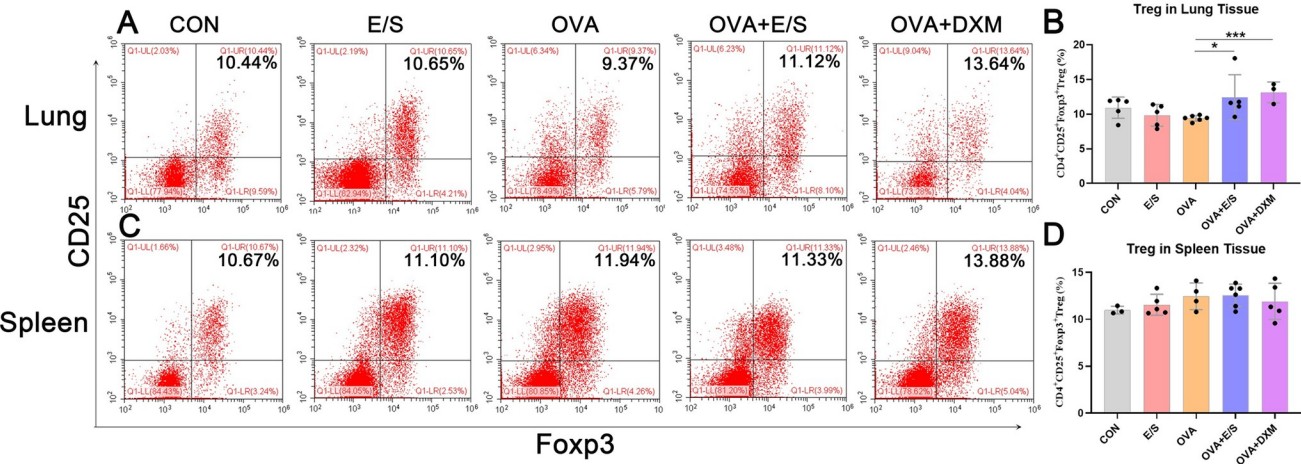

**Fig 4. ESP-SJE treatment upregulated the frequency of Treg in murine lung tissue of AAI model.** (A & B) Representative data of flow cytometry analysis of Treg frequencies (CD4+CD25+Foxp3+ Treg) in lungs (A&B) and spleens (C&D) tissues and their statistical comparisons among CON, E/S, OVA, OVA+E/S and OVA+DXM groups. All data were shown as mean ± SEM. At least 5 mice per group, experiment performed twice. * and *** indicates $P < 0.05$ and $P < 0.001$, ns indicates no significant difference, respectively, by the one-way analysis of variance (ANOVA) with Tukey test.

group (Fig 4A and 4B). In addition, the proportion of Treg in spleen tissue showed a slight increase, although not significant when comparing mice in the OVA+E/S group to those in the OVA group (Fig 4C and 4D).

## ESP-SJEs Treatment Suppressed inflammatory cytokine secretion in BALF

Subsequently, the levels of inflammatory cytokines (i.e., IL-4, IL-5, IL-1β, IL-18, eotaxin, and IL-10) were determined in BALFs using a Luminex assay. As also reported elsewhere [44, 45], OVA sensitization and challenge led to a significant increase in the levels of cytokines IL-4, IL-5, IL-1β, and eotaxin in BALFs compared to mice in the NOR group, whereas the levels of IL-10, and IL-18 were significantly decreased (Fig 5). Furthermore, after treatment with ESP-SJEs during AAI, the levels of IL-4, IL-1β, and eotaxin were significantly reduced in BALFs compared to mice in the OVA group. In addition, IL-10 production in BALFs in the E/S group showed a slight decrease compared to the NOR group, but didn't show any significant difference. Conversely, DXM treatment during AAI significantly reduced the secretion of IL-4, IL-5, IL-1β, IL-18 and eotaxin compared to the OVA group, while the difference in IL-10 secretion was not significant.

## Treg-Inducing Potentially regulatory proteins isolated from ESP-SJEs

Furthermore, we sought to identify potential proteins in ESP-SJEs that might have an effective immunomodulatory activity. ESP-SJEs were separated based on their molecular weight into two different fractions (F1 and F2) by molecular weight, and into five contents (Q1, Q2, Q3, Q4 and Q5) based on their electric charge. ESP-SJEs solutions prepared at different concentrations (5, 10, and 20 mg/mL) were incubated with lymphocytes recovered from the spleen of wide-type mice, and after 3–5 days the proportion of Treg was determined using flow cytometry. As shown in Fig 6A, when ESP-SJEs at a concentration of 10 mg/mL were cultured with lymphocytes, the ability upregulating the proportion of Treg was significant increased compare to other two concentrations. Next, ESP-SJEs fractions at 10 mg/mL were mixed with lymphocytes for 3–5 days, the proportion of Treg in lymphocytes were determined and showed a

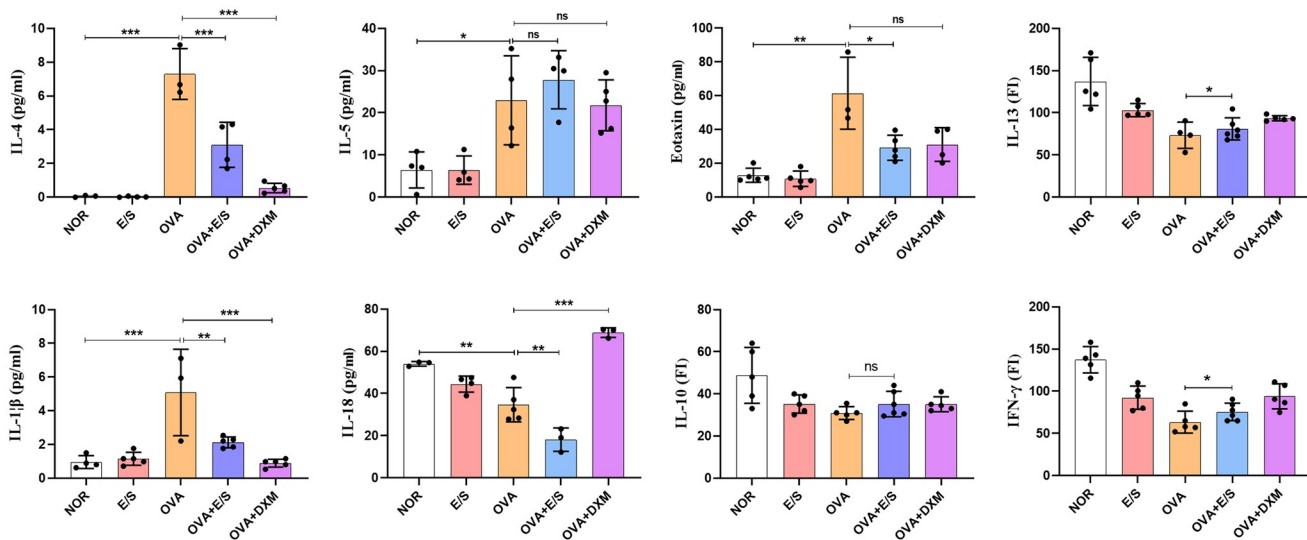

**Fig 5. ESP-SJE treatment affect OVA induced cytokine profile in murine BALF. Concentrations of IL-4, IL-5, IL-13, Eotaxin, IL-1β, IL-18, IL-10 and IFN-γ** in BALF of mice were measured by Luminex between CON, E/S, OVA, OVA+E/S and OVA+DXM groups. All data were shown as mean ± SEM. At least 5 mice per group, experiment performed twice. *, ** and *** indicates $P < 0.05$, $P < 0.01$ and $P < 0.001$, ns indicates no significant difference, respectively, by the one-way analysis of variance (ANOVA) with Tukey test.

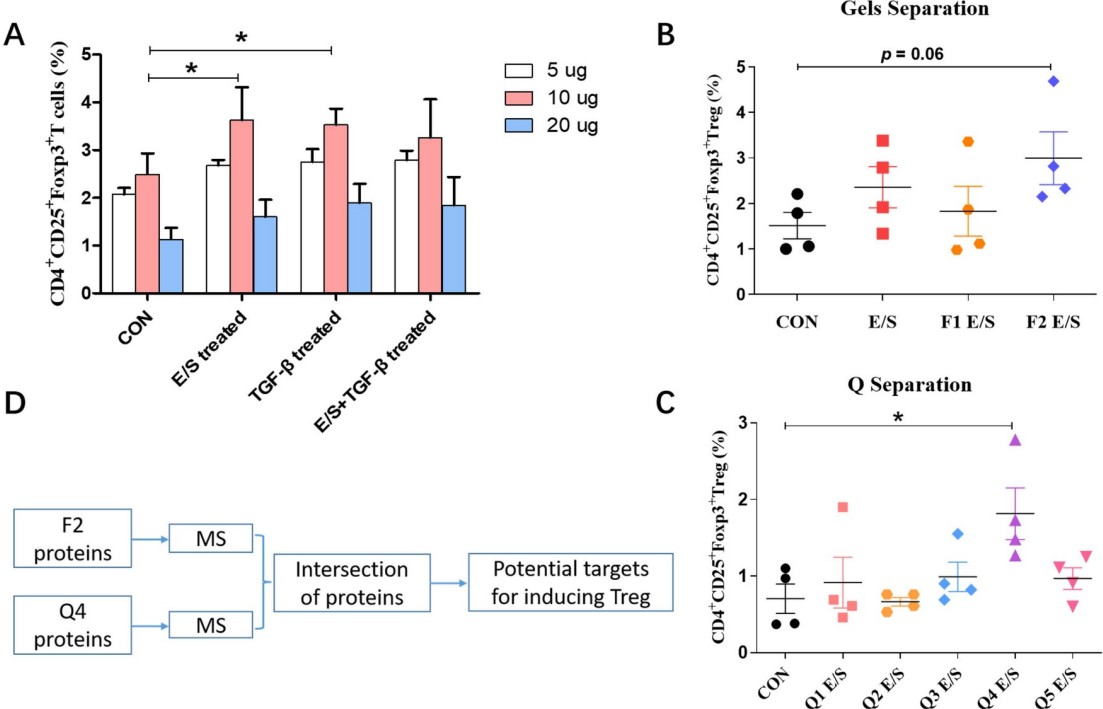

**Fig 6. The ability of Treg (CD4+CD25+FoxP3+) cell induction in murine spleen lymphocyte by ESP-SJE fractions.** (A) The frequency of CD4+CD25+FoxP3+ Treg cell in murine spleen lymphocyte were analyzed with treatment by different E/S concentrations (5, 10 and 20 μg/ml), respectively, between CON, E/S, TGF-β and E/S+TGF-β groups. (B & C) The frequency of CD4+CD25+FoxP3+ Treg cell in murine spleen lymphocyte were analyzed with treatment by CON, E/S, F1 and F2 E/S (two fractions separated by gels separation), and Q1, Q2, Q3, Q4 and Q5 E/S (five fractions separated by Q separation), respectively. (D) The outline of the screened potential modulatory molecules on Treg from ESP-SJE. All data were shown as mean ± SEM. At least 4 mice per group, experiment performed twice. * indicates $P < 0.05$ by the one-way analysis of variance (ANOVA) with Tukey test.

significant upregulation when the lymphocytes were cultured with the fractions F2 and Q4 of ESPs compared to other fractions (Fig 6B and 6C). Moreover, fractions F2 and Q4 were precipitated with tryptases and digested, and the resulting peptides were analyzed via mass spectrometry to identify ESP-SJE-derived peptide sequences (Fig 6D). By matching NCBI and the 4th version of protein database *Schistosoma japonicum* (Sj V4 database), we identified eight ESP-SJEs-derived proteins from F2 and Q4 fractions, whose molecular weight ranged from 4.8–62.1 kDa, and the screened eight proteins showed homology to human proteins with unknown function and identity, which included glutathione S-transferase, egg protein CP422 precursor, tubulin alpha-2/alpha-4 chain, actin-2, T-complex protein 1 subunit beta, T-complex protein 1 subunit epsilon, histone $H_4$, and whey acidic protein core region (Table 1). We further screened the F2 and Q4 fractions for the top three highest expression proteins; in fraction F2, the proteins were glutathione S-transferase, egg protein CP422 precursor, and molecular chaperone HtpG; in fraction Q4, glutathione S-transferase, egg protein CP422 precursor, and T-complex protein 1 subunit beta (Table 1). Total nine potential proteins were primarily identified that might be vital for Treg induction in ESP-SJEs.

## Discussion

Helminth replacement therapy has gained considerable attention in recent years as a form of probiotic therapy. It is emerging as a potential therapeutic source against allergies and dysregulated inflammatory diseases [46]. Among helminth-derived products, excretory/secretory molecules released during helminth infection are indispensable to the protective and immunomodulatory effects in the host [32, 33]. Asthma, one of the most common chronic inflammatory diseases, has been extensively reported to be closely associated with the improvement of sanitary conditions caused by the decrease of infections [3]. It has been shown that helminth infection and helminth-derived ESPs can be effective in alleviating allergic disorders [47–50].

Schistosome was one of the parasites that has been found to have protective effects for allergies like asthma [21, 51, 52]. Report have showed that Schistosome-derived ESPs especially egg-derived ESPs could remarkably modulate host's immune response [53]. However, it was still unknown that whether schistosome egg-derived ESP could alleviate asthmatic inflammation and its potential mechanism. Herein, we sought to investigate the protective roles of ESP-SJEs on AAI in a murine model. The results indicated that ESP-SJEs significantly relieved

**Table 1. Potential proteins for inducing Treg of spleen lymphocyte by ESP-SJE treatment.**

| Accession | Description | Score in Q4[a] | Score in F2[a] | Sequence[b] | MW [kDa][c] |
|---|---|---|---|---|---|
| Sjc_0049920 | Tubulin alpha-2/alpha-4 chain, putative\|Auto - | 274.15 | 101.89 | FDGALNVDLTEFQTNLVPYPR | 48.4 |
| Sjc_0038300 | ko:K00799 glutathione S-transferase [EC2.5.1.18], putative\|Auto - | **217.42** | **108.94** | AEISMLEGAVLDIR | 25.5 |
| Sjc_0002370 | Actin-2, putative\|Auto - | 211.05 | 183.07 | DSYVGDEAQSK | 62.1 |
| Sjc_0200320 | Egg protein CP422 precursor, putative\|Auto - | **163.28** | **49.97** | EGIcVR | 10.3 |
| Sjc_0301360 | ko:K09494 T-complex protein 1 subunit beta, putative\|Auto - | 124.86 | 161.55 | VQDDEVGDGTTSVTVLAAELLR | 57.1 |
| Sjc_0208180 | ko:K09497 T-complex protein 1 subunit epsilon, putative\|Auto - | **93.81** | 93.78 | WVGGPEIELIAIATGGR | 61.1 |
| Sjc_0016100 | Histone H4, putative\|Auto - | 46.96 | 91.26 | DNIQGITKPAIR | 11.4 |
| Sjc_0069680 | IPR008197,Whey acidic protein, core region,domain-containing\|Auto - | 40.73 | 23.74 | cSGcTcGcSccTNcQ | 4.8 |
| Sjc_0044660 | ko:K04079 molecular chaperone HtpG, putative\|Auto - | — | **36.23** | HFSVEGQLEFR | 82.2 |

[a]the similarity of the peptide measurements printed by the mass spectrum to a database of peptides and fragments in the mass spectrum itself.

[b]the amino acid sequence of the peptide measurements printed by the mass spectrum.

[c]the molecular mass of the protein molecule corresponds to the peptide.

OVA-induced AAI by suppressing eosinophil infiltration in the bronchi and goblet hyperplasia, decreasing OVA-specific IgE, IgG1 and IgG2a levels, as well as inhibiting the secretion of Th2 cytokines IL-4, IL-5 and chemokine eotaxin. These results were consistent with previous findings that chronic *S. mansoni* infection had a protective effect on AAI in mice [23], although they do not confirm whether the protective effect was due to eggs or to egg-derived factors.

Treg cells play a crucial role in controlling asthmatic pathogenesis by exerting immunosuppressive functions [54, 55]. It has been reported that depleting Treg cells could aggravate inflammation in a mouse model of asthma, whereas the transfer of Treg cells had a modulatory effect [56–58]. Several studies indicated that schistosome-mediated suppression of AAI was dependent on the presence of Treg cells induced by viable eggs and infection in a murine model of asthma [21, 44], which the results was consistent with our results that ESP-SJE could significantly upregulate the frequency of Treg especially the Treg of lung-tissue. In recent study using a mouse model, we also found that lung-stage *S. japonicum* infection established a regulatory environment in the lungs mediated by Treg, which helped to relieve OVA-induced AAI [59]. For *S. mansoni* and asthma, other studies have also reported that the presence of Foxp3[+] Treg cells was necessary for the helminth-mediated suppression of AAI [44, 47]. Collectively, these results suggested that Treg cells induced by schistosomes at different developmental stages were the major suppressive mechanism against the host's immune disorders. Among other helminths, Ebner et al. reported that proteins released from *T. suis* larvae could efficiently dampen allergic airway hyperreactivity in an animal model through the activity of anti-inflammatory cytokine IL-10 [30], which contradicted the results of the present study. These discrepancies suggested that different helminths may employ different regulatory mechanisms against the host's disordered immune system.

During schistosome infection, the egg-laying stage is a vital period in the host-parasite dynamic development, influencing the host's immune response phenotype and the stimuli driving pathological conditions. In the early phase (before 6 weeks), juvenile and adult schistosomes evade the host's immune response, while in the later phase (after 6 weeks), schistosome eggs secrete immunomodulatory proteins that manipulate the host's immune response to promote egg/host survival for a prolonged period [60]. A previous study has indicated that E/S antigens derived from *S. japonicum* eggs can drive macrophages polarization into the immunomodulatory M2b phenotype [33]. These findings showed that ESP-SJEs contain abundant anti-inflammatory and modulatory molecules that affect the host's immune response. Another study reported that AAI induced by house dust mite was significantly relieved after *S. japonicum* infection at egg-laid phase [21]. These results support the hypothesis of the present study that ESP-SJEs can modulate OVA-induced AAI. It has been reported that *S. mansoni* infection at the chronic induced suppressive immune reaction to inhaled allergens [23], thus suggesting a similar immune effect between modulatory molecules released by *S. mansoni* and *S. japonicum*. However, the molecular mechanism by which ESP-SJEs protect against allergic disorders remains to be further explored.

Asthma is typically characterized with Th2-type cytokines production and eosinophils-dominant chronic airway inflammation. Th2 cytokines IL-4, on one hand, can activate epithelial and endothelial cells to produce eotaxin, a chemotaxin important for recruiting eosinophil from the airway microvessels into the lung tissue [61], on the other hand, facilitate the antibody IgE production, which mediate the secretion of inflammatory medium [62]. In present study, we found that ESP-SJEs significant modulate OVA-induced IL-4 and eotaxin response, along with the inhibition of the antibody of IgE and airway inflammation. In addition, we found OVA sensitization and challenge also significantly upregulate IL-1β secretion, which IL-1β-mediated inflammation could recruit inflammatory cells, especially inflammatory

monocytes/macrophage after OVA/alum sensitization [63], which this result just acts in cooperation with Fig 1B that OVA-induced macrophage inflammation was the second biggest infiltration inflammatory cell besides the eosinophils.

We tried to screen the active fractions or molecules from ESP-SJEs, which could modulate the inflammation response of AAI. First, we performed an optimal concentration of ESP-SJE as 10 μg/ml to better induce Treg *in vitro*. Then, ESP-SJEs were separated into different fractions using gel separation and an ion-exchange column, respectively. The ability of Treg induction were performed by different ESP-SJEs fractions *in vitro* and the results showed that two fractions F2 and Q4 groups were the best inducer for Treg. So, the two fractions were performed by LC-MS analysis and the proteins both from F2 fraction and from Q4 fraction and the top three high-expressed were regarded as potential inducing-Treg molecules. A total of nine potential proteins were isolated and identified from ESP-SJEs including Sjc_0049920, Sjc_0038300, Sjc_0002370, Sjc_0200320, Sjc_0301360, Sjc_0208180, Sjc_0016100, Sjc_0069680, and Sjc_0044660, which included two cytoskeleton proteins (tubulin and actin-2); two enzyme-like molecules (glutathione S-transferase and the molecular chaperone HtpG with histidine kinase-like ATPases); an uncharacterized schistosome protein (egg protein CP422 precursor); two functional protein T-complex protein 1 related molecules; and histone $H_4$ and whey acidic protein. Based on our results, we hypothesized that these functional or enzyme-like proteins may play an important immunoregulatory role in OVA-induced AAI by inducing Treg cells. Furthermore, these proteins may constitute potential candidates for the treatment of allergic disorders and autoimmune diseases, which highlights the importance of continuing research on helminth-derived molecules that have protective effect against such disorders. Future studies should encompass recombinant expression of the identified proteins to elucidate their individual immunomodulatory activity in order to refine helminth-derived therapeutics.

## Conclusion

In the present study, this was the first-hand evidence that ESP-SJEs could effectively alleviate OVA-induced AAI in a mouse model by the induction of Treg cells in lung tissue. And we screened nine potential proteins molecules from ESP-SJEs which may be the best inducer for Treg to inhibit AAI. However, in present study, the roles of potential proteins inducing Treg on AAI did not get verified by examination. Nonetheless, we think that it is very likely the observed therapeutic effect was a collective result of multiple components of schistosome, as previous studies showed multiple enzymes released by schistosome egg-derived molecules could regulate host immunity [53]. We plan to acutely define these components in future. Further, the underlying mechanisms of which these potential proteins modulate OVA-induced AAI and the role of Treg during the process require further investigation. Thus, identifying the active fractions or molecules in ESP-SJEs is of particular importance, since it would facilitate the feasibility of helminth therapy in future clinic.

## Supporting information

**S1 Data. Excel spreadsheet containing, in separate sheets, the underlying numerical data and statistical analysis for Figs 1, 2, 3, 4, 5 and 6 and their panels.**
(XLSX)

**S1 Fig. The overview images (10 ×) of H&E staining of lung tissue among CON, E/S, OVA, OVA+E/S and OVA+DXM groups, at least 5 individuals in each group.** CON, normal mice (without OVA sensitization and challenge); E/S, mice without OVA sensitization and

challenge but treated with ESP-SJE; OVA, mice with OVA sensitization and challenge but without ESP-SJE treatment; OVA + E/S, mice sensitized and challenged with OVA and treated with ESP-SJE; OVA + DXM, mice sensitized and challenged with OVA and treated with dexamethasone.
(TIF)

**S2 Fig. The overview images (10 ×) of PAS staining of lung tissue among CON, E/S, OVA, OVA+E/S and OVA+DXM groups, at least 5 individuals in each group.**
(TIF)

**S3 Fig. The overview data of flow cytometry analysis of Treg frequencies (CD4+CD25 +Foxp3+ Treg) in lungs tissues among CON, E/S, OVA, OVA+E/S and OVA+DXM groups, 3–6 different individuals in each group.**
(TIF)

**S4 Fig. The overview data of flow cytometry analysis of Treg frequencies (CD4+CD25 +Foxp3+ Treg) in spleen tissues among CON, E/S, OVA, OVA+E/S and OVA+DXM groups, 3–6 different individuals in each group.**
(TIF)

**S5 Fig. The gate strategy of CD4$^+$CD25$^+$Foxp3$^+$Treg of lymphocytes of spleen or lung tissues showed by flow cytometry.**
(TIF)

**S6 Fig. The proportions of CD4$^+$CD25$^+$Foxp3$^+$Treg of spleen lymphocytes from 5 individual mice including control, 5 μg/ml ESP-SJEs treated, 5 μg/ml TGF-β treated, and 5 μg/ml ESP-SJEs plus TGF-β treated groups, respectively.**
(TIF)

**S7 Fig. The proportions of CD4$^+$CD25$^+$Foxp3$^+$Treg of spleen lymphocytes from 5 individual mice including control, 10 μg/ml ESP-SJEs treated, 10 μg/ml TGF-β treated, and 10 μg/ml ESP-SJEs plus TGF-β treated groups, respectively.**
(TIF)

**S8 Fig. The proportions of CD4$^+$CD25$^+$Foxp3$^+$Treg of spleen lymphocytes from 5 individual mice including control, 20 μg/ml ESP-SJEs treated, 20 μg/ml TGF-β treated, and 20 μg/ml ESP-SJEs plus TGF-β treated groups, respectively.**
(TIF)

## Acknowledgments

We appreciate the linguistic assistance provided by TopEdit (www.topeditsci.com) during the preparation of this manuscript.

## Author Contributions

**Conceptualization:** Zhidan Li.

**Data curation:** Zhidan Li, Xiaoling Wang, Wei Zhang.

**Formal analysis:** Zhidan Li, Xiaoling Wang, Wenbin Yang.

**Funding acquisition:** Wei Hu.

**Methodology:** Zhidan Li, Wei Zhang.

**Software:** Zhidan Li, Bin Xu.

**Supervision:** Wei Hu.

**Validation:** Zhidan Li.

**Writing – original draft:** Zhidan Li.

**Writing – review & editing:** Wei Hu.

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
