## [Decision Letter · Decision Letter 0]

22 Jun 2023

Dear Professor Hu,

Thank you very much for submitting your manuscript "Excretory/Secretory Products from Schistosoma japonicum  Eggs Alleviate Ovalbumin-Induced Allergic Airway Inflammation" for consideration at PLOS Neglected Tropical Diseases. As with all papers reviewed by the journal, your manuscript was reviewed by members of the editorial board and by several independent reviewers. In light of the reviews (below this email), we would like to invite the resubmission of a significantly-revised version that takes into account the reviewers' comments. 

The ms will be considered for publication provided the authors revise the ms according to the comments of the reviewers .

We cannot make any decision about publication until we have seen the revised manuscript and your response to the reviewers' comments. Your revised manuscript is also likely to be sent to reviewers for further evaluation.

Sincerely,

Zvi Bentwich, M.D

Academic Editor

Cinzia Cantacessi

Section Editor

The ms will be considered for publication provided the authors revise the ms according to the comments of the reviewers .

Reviewer's Responses to Questions

**Key Review Criteria Required for Acceptance?**

**Methods**

-Are the objectives of the study clearly articulated with a clear testable hypothesis stated?

-Is the study design appropriate to address the stated objectives?

-Is the population clearly described and appropriate for the hypothesis being tested?

-Is the sample size sufficient to ensure adequate power to address the hypothesis being tested?

-Were correct statistical analysis used to support conclusions?

-Are there concerns about ethical or regulatory requirements being met?

Reviewer #1: The authors clearly stated the objective of the study and have obtained permission from the appropriate ethic regulatory bodies for the study.

Reviewer #2: - It is unclear how many experiments were performed and how many mice were included in the groups or how many different spleens were used to do the in vitro experiments. Please add to the legend and show all individual mice (separate data points) from all experiments in the graphs of the figs. 

- When more than two groups are included in the experiments, ANOVA should be applied with post hoc analysis between groups. This may have been applied, but this is not clearly described in the methods nor the legends. This should be adapted.

- The description of the legends for the suppl. figures is missing. Please add.

**Results**

-Does the analysis presented match the analysis plan?

-Are the results clearly and completely presented?

-Are the figures (Tables, Images) of sufficient quality for clarity?

Reviewer #1: The results are well presented. However, the image quality and labels in the manuscript are not of quality. I kindly suggest that authors improve on this.

Reviewer #2: - In figure 1, the y-axis is staggered, which makes it difficult to interpret the data. Please use a continuous axis.

- In suppl fig 1; three displays are presented for the the different concentration of ES/P - please combine in one graph to allow full comparison between the groups. The percentages of Treg cells seem to differ quite substantially between the three displays; this makes it hard to interpret the data and also questions the scientific rigor of the findings.

- Why does the TGFb not induce Treg cells? The positive control does not seem to be working?

- In the text is the described that also the F2 and Q4 induce more Treg cells in Suppl fig 2 (lines 412-414)- but this shows something else. These data seem to be lacking. Please add.

- The data in Suppl fig 1 and 2 seems very important for the message and conclusions of the paper and should be part of the main figures. If not sufficient experiments are performed to merit a main figure, please add more data to allow this.

**Conclusions**

-Are the conclusions supported by the data presented?

-Are the limitations of analysis clearly described?

-Do the authors discuss how these data can be helpful to advance our understanding of the topic under study?

-Is public health relevance addressed?

Reviewer #1: • The authors provided conclusions which are supported by the data presented in the manuscript.

• The limitations of the study have not been mentioned. I suggest authors provide information on the limitation of their study in the manuscript.

• The benefit of the findings to the advancement of knowledge on the topic under study have been discussed and public health relevance has also been touched on.

Reviewer #2: The data show significant, but modest changes. This is not always reflected by the accompanying text. In particular, the abstract needs to be tuned down, but also in other sections of the manuscript.

Fox example:

- line 39: ..reducing the number of eosinophils in the lungs, but not in the BAL ... (differences between different compartments)

- line 40: ...secretion of inflammatory cytokines, like ... (not all were reduced)

- line 41: ... a significant, but modest, upregulation of Treg cells ... (Differences were very modest)

- line 42-44: this conclusion is exaggerated - some of the fractions may induce a modest induction of Tregs AND several molecules were identified. But it is not tested whether those 9 indeed induce Treg cells - the way it is written now, may suggest it is. Please adapt.

- line 49-50: Also here the conclusion that inhibition of AAI is likely due to Treg cell induction is too much; this was not tested. Here only can be included that there is a modest inhibition of AAI combined with a modest upregulation of Treg cells. Whether this is linked has not been tested.

- line 52: Also the conclusion that ES/P molecules can be used for therapeutic application is too strong - at most they are likely candidates to be tested for their therapeutic application. 

These conclusions should also be tuned down in other sections of the manuscript.

The differences in cytokine expression and the biological relevance has not been discussed in the discussion section.

I miss an strength and limitation section in the discussion.

**Editorial and Data Presentation Modifications?**

Reviewer #1: Minor Revision

Reviewer #2: See comments above

**Summary and General Comments**

Reviewer #1: • The authors did not present the abstract in a format recommended by the journal. I suggest the authors follow the journal recommendation and present the abstract in sections with headings

• There is absence of author’s summary in the manuscript as recommended by the journal. I suggest authors write on “AUTHOR’S SUMMARY” and this information should come after the “ABSTRACT”

• Line 169: Remove the word “of” in between “with” and “ESP-SJEs”

• Line 169: The bracketed information “(1 mg each mouse each time)” is not clear. I suggest that this information be presented as “(1mg per mouse per time)”

• Line 256: Remove the word “using” in between “chromatography” and “with”

• Line 294: Kindly provide space in between the combined word “spectrometercoupled”

Reviewer #2: See comments above.

PLOS authors have the option to publish the peer review history of their article (what does this mean?). If published, this will include your full peer review and any attached files.

Reviewer #1: No

Reviewer #2: Yes: Hermelijn H. Smits
---

## [Editor Report · Decision Letter 1]

29 Aug 2023

Dear Professor Hu,

We are pleased to inform you that your manuscript 'Excretory/Secretory Products from Schistosoma japonicum  Eggs Alleviate Ovalbumin-Induced Allergic Airway Inflammation' has been provisionally accepted for publication in PLOS Neglected Tropical Diseases.

Best regards,

Zvi Bentwich, M.D

Academic Editor

Cinzia Cantacessi

Section Editor

---

## [Editor Report · Acceptance letter]

19 Sep 2023

Dear Professor Hu,

We are delighted to inform you that your manuscript, "Excretory/Secretory Products from Schistosoma japonicum  Eggs Alleviate Ovalbumin-Induced Allergic Airway Inflammation," has been formally accepted for publication in PLOS Neglected Tropical Diseases.

Best regards,

Shaden Kamhawi

co-Editor-in-Chief

Paul Brindley

co-Editor-in-Chief
